# Blood-Brain Barrier-Associated Proteins Are Elevated in Serum of Epilepsy Patients

**DOI:** 10.3390/cells12030368

**Published:** 2023-01-19

**Authors:** Elżbieta Bronisz, Agnieszka Cudna, Aleksandra Wierzbicka, Iwona Kurkowska-Jastrzębska

**Affiliations:** 1Second Department of Neurology, Institute of Psychiatry and Neurology, 02-957 Warsaw, Poland; 2Sleep Disorders Center, Department of Clinical Neurophysiology, Institute of Psychiatry and Neurology, 02-957 Warsaw, Poland

**Keywords:** MMP-9, MMP-2, TIMP-1, TIMP-2, S100B, blood–brain barrier, inflammation, ECM, epilepsy, seizure

## Abstract

Blood–brain barrier (BBB) dysfunction emerges as one of the mechanisms underlying the induction of seizures and epileptogenesis. There is growing evidence that seizures also affect BBB, yet only scarce data is available regarding serum levels of BBB-associated proteins in chronic epilepsy. In this study, we aimed to assess serum levels of molecules associated with BBB in patients with epilepsy in the interictal period. Serum levels of MMP-9, MMP-2, TIMP-1, TIMP-2, S100B, CCL-2, ICAM-1, P-selectin, and TSP-2 were examined in a group of 100 patients who were seizure-free for a minimum of seven days and analyzed by ELISA. The results were compared with an age- and sex-matched control group. Serum levels of MMP-9, MMP-2, TIMP-1, TIMP-2 and S100B were higher in patients with epilepsy in comparison to control group (*p* < 0.0001; <0.0001; 0.001; <0.0001; <0.0001, respectively). Levels of CCL-2, ICAM-1, P-selectin and TSP-2 did not differ between the two groups. Serum levels of MMP-9, MMP-2, TIMP-1, TIMP-2 and S100B are elevated in patients with epilepsy in the interictal period, which suggests chronic processes of BBB disruption and restoration. The pathological process initiating epilepsy, in addition to seizures, is probably the factor contributing to the elevation of serum levels of the examined molecules.

## 1. Introduction

Epilepsy is a chronic neurological disorder affecting 1% of the world population and the most common neurological disease [1]. Causes of epilepsy comprise a variety of structural and functional changes of the central nervous system (CNS). Moreover, seizures can also occur in the course of other diseases. Diverse mechanisms underlie the initiation of seizures and development of epilepsy, and although epilepsy has been researched for a long time, new hypotheses regarding its pathogenesis are still emerging.

There is growing evidence indicating the importance of BBB dysfunction in the development of seizures and epilepsy. Increased permeability of BBB plays an important role in brain damage and leads to the modification of the extracellular matrix (ECM) [2], influx of blood cells, endo and exogenic substances and pathogens, causing local inflammation and angiogenesis [3]. Proinflammatory mediators increase even further BBB permeability in a positive feedback loop [4]. In the situation of dysfunctional BBB, endothelial cells are activated and interact with circulating leukocytes, which results in leukocyte transmigration through BBB and the following generation of seizures [3,5]. Long-lasting BBB disruption leads to the development of epilepsy—epileptogenesis, causing chronic recurrent epileptic seizures [6].

Regulation of BBB permeability is complex and depends not only on the state of cells that comprise BBB (endothelial cells, pericytes, astrocytic end-feet), but also neighboring and circulating molecules and cells (local and circulating mediators of inflammation, local neurons, myocytes, astrocytes and ECM, blood-borne leukocytes, etc.) [7,8,9,10]. Matrix metalloproteinases (MMPs) modify ECM, modulate inflammation and increase BBB permeability [9,11]. Increased activity of MMPs was observed in epileptogenic zones in patients with focal cortical dysplasia (FCD) [12] and tuberous sclerosis (TSC) [13], two types of epilepsy characterized with high activity of disease. Serum level of MMP-9 was elevated after generalized tonic-clonic seizures (GTCS) [14]. Tissue inhibitors of metalloproteinases (TIMPs) counteract the actions of MMPs and are associated with neuroprotection [15]. Their serum level is increased after GTCS [16,17]. TSP-2 is another protein that plays a restorative role, able to non-specifically inhibit MMPs. Loss of TSP-2 is associated with slower BBB repair and intensification of inflammation [18]. Patients with temporal lobe epilepsy (TLE) had higher serum levels of TSP-2 than the control group [19]. Serum S100B is known as a biomarker of brain damage [20] as it correlates with the albumin quotient. In high concentrations, S100B exacerbates inflammation [21]. S100B is overexpressed in TLE [22] and its level in serum is elevated after seizures [23,24]. Another proinflammatory protein is CCL-2, which participates in leukocyte adhesion [25], modifies ECM [26] and induces microglia [27]. CCL-2 is increased after seizures [28]. ICAM-1 and P-selectin (P-sel) are molecules that participate in leukocyte rolling and binding [29,30], thus enabling leukocyte trafficking through BBB. Serum levels of these molecules were elevated in patients with status epilepticus [31], which might suggest increased BBB permeability and intensification of leukocyte influx to the CNS.

A set of molecules associated with BBB permeability (MMP-2, MMP-9, S100B), restoration (TIMP-1, TIMP-2, TSP-2), neuroinflammation (CCL-2), and endothelial activation (ICAM-1, P-sel) that are affected in epilepsy and can be measured in blood was selected in this study. The aim of the present study was to evaluate serum concentrations of BBB-associated proteins in patients with epilepsy in the interictal period with a control group.

## 2. Materials and Methods

The patient group consisted of 100 successive adult patients with epilepsy who attended the outpatient clinic of the Institute of Psychiatry and Neurology in Warsaw, Poland. Strict inclusion and exclusion criteria were applied for the recruitment of the study participants (Table 1). We included patients after a 7-day period of seizure freedom based on our previous results [24,32]. The exclusion criteria included factors which could influence the examined markers with the exception of disease activity, which did not affect qualification to the study. The same exclusion criteria were applied for control group. The qualification took place in the period from 20 July 2015 to 21 July 2017. All the patients confirmed their willingness to participate in the study by signing informed consent.

Detailed characteristics of patients were obtained, including demographic data, etiology of epilepsy, type of seizures, age of epilepsy onset, disease duration, anti-seizure medication, and time of seizure freedom. The etiology of epilepsy (genetic, structural-metabolic, unknown) was diagnosed according to the classification proposed by ILAE [34].

Patients were recommended to fast, refrain from smoking, using alcohol and undertaking vigorous physical activities for 24 h before the examination. At the day of examination, between 8 and 9 a.m., blood was collected; afterwards, we performed standard EEG in a 10–20 montage for 30 min. To ascertain similar conditions, patients were asked to rest for 5 to 15 min before blood collection. Samples were centrifuged and the supernatant was divided into two portions. One portion was examined with standard laboratory tests for CRP and the second one was frozen and stored in closed test tubes at −80 °C for further analysis, when it was thawed before the immunoenzymatic testing.

We analyzed a panel of nine BBB-associated proteins (MMP-2, MMP-9, TIMP-1, TIMP-2, S100B, TSP-2, CCL-2, ICAM-1, P-sel) using sandwich-type ELISA kits following the manufacturers’ instructions. For the S100B kit, the producer was Merck Millipore, Darmstadt, Germany, and all the remaining proteins were analyzed using kits by R&D Systems, Minneapolis, MN, USA. Multiskan Go microplate reader (Thermo Scientific, Waltham, MA, USA) was used to acquire data from the reactions. Calculations of protein concentrations were carried out following the manufacturers’ instructions.

Preliminarily, descriptive statistics (means, standard deviation, percentiles, and correlations) were used to analyze the data. Depending on the distributions of probability, one-dimensional calculations were performed using Student’s or Wilcoxon’s tests for continuous variables, and chi-square test or Fisher’s exact test were used for nominal variables. We used the generalized linear mixed models (GLMMs) to identify variables influencing the serum levels of BBB-associated proteins. For the optimal model selection, Akaike information criterion (AIC) statistics were used, as this method is asymptotically equivalent to resampling methods [35]. Model selection was performed according to the practical information–theoretic approach [36]. The majority of the variables were modeled with the assumption of normal distribution, gamma distribution was optimal in few cases and variables with over 10% of zero values were analyzed using the Tweedie distribution. SAS package (SAS/STAT rel. 15.1) was used for calculations. The *p* value of 0.05 was considered statistically significant. There was no missing data. We performed the data analysis from 5 May 2016 to 13 August 2021.

The preparation of the manuscript followed the STROBE checklist [37].

## 3. Results

### 3.1. Research Group

The research group consisted primarily of 115 patients, 15 of which were excluded (see Figure 1), yielding 100 patients, 52 women and 48 men. The mean age of the patients was 43.01 ± 1.53 years, and the median was 39 years. There were no differences in age between men and women. All 100 patients completed follow-up and, afterwards, were analyzed in the study. The patients were examined after a minimum 7-day period of seizure freedom. Mean duration of seizure freedom was 432.64 ± 66.50 days, median was 140 days (range 7–4036 days) and 35% of patients did not experience seizures during the last year.

As far as epilepsy etiology is concerned, there were 35 patients with genetic epilepsy (juvenile myoclonic epilepsy and idiopathic genetic epilepsy), 50 patients with structural-metabolic epilepsy and 15 patients with epilepsy of unknown etiology. In the group of patients with structural-metabolic epilepsy, the largest subgroup were patients with post-traumatic epilepsy (14 patients) and post-stroke epilepsy (12 patients). The remaining subgroups were 10 patients with mesial temporal lobe epilepsy (MTLE), two of whom had confirmed hippocampal sclerosis (HS), six patients with neonatal hypoxic-ischemic encephalopathy, five patients with congenital malformations of CNS, and three patients with epilepsy developed after CNS infection.

A total of 45 patients had GTCS (both primary and secondary), 17 patients had focal motor seizures and 38 patients had both focal seizures and secondary GTCS. EEG examination carried out after blood collection was normal in 45% of patients. In 17%, abnormal background activity was observed. Focal slow waves were recorded in 49% of patients and 17% had epileptiform discharges without clinical manifestation. Out of 100 patients, 99 were treated and one patient had newly diagnosed epilepsy and was analyzed before introduction of medication. A total of 65 patients were on monotherapy and 34 patients needed polytherapy (31 with two, and three with three drugs).

The demographic characteristics of the research and control group are shown in Table 2.

### 3.2. Control Group

The control group consisted of 100 sex- and age-matched subjects, 52 women and 48 men, and did not differ from research group in terms of sex and age. The mean age was 42.16 ± 1.58 years, the median 39 years, min–max was 17–84 years. There were no differences in age between the men and women in control group.

### 3.3. Serum Levels of BBB-Associated Proteins

Serum levels of MMP-9, MMP-2, TIMP-1, TIMP-2, S100B, proteins associated with increased BBB permeability, were higher in the research group than in the control group (Table 3). MMP-2/TIMP-2 ratio was also higher in the research group. Serum levels of the proteins associated with activation of endothelium (P-sel, ICAM-1), inflammation (CCL-2), and synaptogenesis (TSP-2) did not differ between the two groups.

### 3.4. Influence of Demographic Factors on the Serum Levels of BBB-Associated Proteins

We did not observe significant correlations between age and serum levels of BBB-associated proteins in the research group. Interestingly, there was a weak positive correlation between age and serum CCL-2 level in the control group (r = 0.24, *p* = 0.0151; Figure 2).

Serum levels of TIMP-2 were higher in women both in the research and in the control group. Mean serum levels of TIMP-2 in women with epilepsy were 155.91 ± 5.52 ng/mL, median 149.97 ng/mL vs. 142.13 ± 5.46 ng/mL and 133.84 ng/mL in men (*p* = 0.0172). In the control group, mean serum levels of TIMP-2 in women were 128.03 ± 4.93 ng/mL, median 124.84 ng/mL and in men 111.54 ± 5.03 ng/mL, median 111.14 ng/mL (*p* = 0.0015; Figure 3).

The GLMMs confirmed the sex-related differences in TIMP-2 levels in the research group (least squares means; women: 155.30 ± 4.65 ng/mL, men: 143.91 ± 4.53 ng/mL, *p* = 0.0397). Additionally, higher levels of S100B were observed in men. Mean level of S100B in men was 77.62 ± 13.13 pg/mL, in women 40.89 ± 6.58 pg/mL (*p* = 0.0061, least squares means).

### 3.5. Correlation between the Serum Levels of BBB-Associated Proteins

There was a strong positive correlation between the serum level of MMP-9 and MMP-9/TIMP-1 ratio, and between the level of MMP-2 and MMP-2/TIMP-2 ratio. The level of TIMP-1 showed a strong negative correlation with the MMP-9/TIMP-1 ratio. A moderate positive correlation was observed between the levels of MMP-9 and P-selectin as well as between levels of MMP-2 and TIMP-2. The level of MMP-2 showed a moderate negative correlation with the level of TSP-2. Remaining correlations between the examined proteins were weaker (Table 4).

## 4. Discussion

### 4.1. Proteins Associated with BBB Disruption and Restoration—MMP-9, MMP-2, TIMP-1, TIMP-2, S100B

We observed higher serum levels of MMP-9, MMP-2, TIMP-1, TIMP-2, and S100B in the research group in comparison to the control group. The examination took place after a 7-day period of seizure freedom to exclude the influence of recent seizures on serum levels of the examined proteins. Elevation of serum level of S100B, which is a well-known biomarker of BBB activation/disruption, suggests that this process is present in patients with epilepsy in the interictal period. Increased activity of MMP-9 [38], overexpression of MMP-2 [12], TIMP-1 [39], TIMP-2 [38], and S100B [22] were found within the epileptogenic zone where repeating epileptic discharges lead to BBB dysfunction. In our study, we showed that these proteins can also be measured in serum and that their levels are higher in patients with epilepsy, which might indicate chronic BBB dysfunction.

#### 4.1.1. Short Characteristics of Proteins Associated with BBB Disruption and Restoration

MMPs are proteolytic enzymes responsible for ECM degradation and an increase in BBB permeability. MMPs participate in various physiological processes, such as cell differentiation and migration, tissue remodeling, cytokine secretion, regulation of trophic factors and maintaining balance between pro- and anti-inflammatory factors [11,40]. They also play an important role in neuroinflammation, neurotoxicity, carcinogenesis and lysis of BBB basal lamina [41,42].

MMP-9, also known as gelatinase B, is a complex protein able to bind a variety of substrates such as collagen type I and IV [43], chemokines [44], interleukins [45], precursors of growth factors [46], tight junctions [47], synaptic proteins [48], and TIMPs. MMP-9-dependent lysis of dystroglycan facilitates leukocyte transmigration through ECM [49] and exacerbates local inflammation. MMP-9 modifies synaptic transmission through changes in ECM, and modulation of glutamatergic transmission by the increasing activity of NMDAR [50] and decreasing the activity of AMPAR [51].

MMP-2 (gelatinase A) is another MMP associated with BBB dysfunction. Similarly to MMP-9, MMP-2 degrades BBB basal lamina and tight junctions [52]. Other substrates of MMP-2 are ECM components, such as laminins, aggrecan, collagen [40], growth factors [53], and IL-1β [54]. An important difference between MMP-2 and MMP-9 lies in their expression. In physiological conditions, expression of MMP-9 is low, but it rises significantly after neuronal depolarization and receptor activation [55,56], and in the presence of proinflammatory factors [57]. MMP-2 is expressed constitutively at higher levels than MMP-9 [58,59], and inflammatory mediators cause a small increase in its expression [60].

TIMPs act as antagonists to MMPs, with TIMP-1 binding preferentially with MMP-9 and TIMP-2 with MMP-2. TIMPs prevent excessive cell damage and cell death [61], influence cell differentiation, growth and migration [61,62,63], impact activation of growth factors [64] and regulate long-term potentiation [65]. Higher expression of TIMPs is associated with lower BBB disruption, which suggests a neuroprotective role of TIMPs within the CNS [66]. The balance between MMPs and TIMPs is essential for maintaining CNS in health [67].

S100B is a calcium-binding protein known as a biomarker of BBB disruption [68]. S100B influences cell proliferation, differentiation and growth [69,70], calcium homeostasis [71] and enzyme activity within the cell [72]. Low amount of S100B is secreted in a constitutive manner and its secretion is increased after CNS damage [73]. S100B acts differently depending on its concentration. At low concentration, it might be associated with increased permeability of BBB without CNS damage [20] and promote nerve growth [74], neuronal survival [75], and long-term potentiation [76]. High concentration of S100B is correlated with neurotoxicity and inflammation [77,78] mediated by binding with RAGE [21], activation of microglia and astroglia and increased expression of proinflammatory mediators [79,80].

#### 4.1.2. Proteins Associated with BBB Disruption and Restoration in Epilepsy

There is growing evidence on the BBB-associated proteins in animal models and specimens obtained from patients during epilepsy surgery. Higher MMP-9 expression in the hippocampus correlates with increased susceptibility to seizures [81] and BBB permeability [82]. In a mouse model of PTZ-induced kindling, degradation of one of ECM components, perineuronal nets, through MMPs was associated with an increased number of seizures [83]. In a rat model of electrically induced epilepsy, increased expression of MMP-2 was found a week after seizures [84].

MMP-9 activity was higher in surgical specimens of patients with MTLE with HS [38], FCD [12] and TSC [13]. Increased MMP-2 expression was found in surgical specimens of adult patients with FCD, but not in children [12].

MMP-9 was increased after GTCS in CSF [85] and serum [14,23]. Serum MMP-9 was also showed to be increased in a general population of patients with epilepsy [86,87], but the precise time from the last seizure was either not mentioned or fitted in a broad range, starting from 0 days. Increased serum level of MMP-9 was found in patients with HS, yet the period of seizure freedom before examination was also unknown and it cannot be excluded that recent seizures influenced these results [88]. Our study shows that the serum level of MMP-9 is increased in patients with epilepsy even without recent seizures, which suggests chronic BBB disruption and ECM alterations in patients with epilepsy.

The study examining CSF MMP-2 after GTCS showed no differences between the level of MMP-2 in the research group and the control group [85]; yet, the control group consisted of patients diagnosed due do other neurologic diseases. There were no statistically significant differences in serum MMP-2 at 1 h after GTCS (although a tendency for higher levels of MMP-2 could be observed), and at 72 h serum MMP-2 was lower than in the control group [16]. In patients with epilepsy examined regardless of the time from last seizure, the serum level of MMP-2 was lower than in control group [89,90,91]. In our study, serum MMP-2 level was higher in patients with epilepsy. As the results are contradictory, further research is needed. The differences might be a result of examining the serum at different time points from last seizure.

In a rat model of electrically induced epilepsy, authors observed increased expression of TIMP-1 and TIMP-2 [84]. A sequential increase in TIMP-1 expression, first in neurons and after 3 days in astrocytes, was showed in a model of kainate-induced epilepsy [92]. In this model, TIMP-1 activity was correlated with neuroprotection. Increased expression of TIMP-2 in microglia was found in dogs with epilepsy [93]. In patients with MTLE with HS, low TIMP-1 activity was found in CA1 and CA2 and high TIMP-1 activity was observed in the neocortex [38]. In the same study, moderate TIMP-2 activity was found in CA1 and high TIMP-2 activity in the neocortex and mossy fibers [38]. Increased levels of TIMP-2 was observed in adult patients with FCD [12].

The serum level of TIMP-1 was higher in children with febrile seizures in course of HHV infection than in children without febrile seizures [94], but lower in children with prolonged febrile seizures with encephalopathy than in children with febrile seizures without encephalopathy, status epilepticus, West syndrome, and control group [95]. A study of serum TIMP-1 kinetics after GTCS showed its increase [17]. In our study, patients examined in the interictal period had increased serum levels of TIMP-1, which suggests long-term activation of BBB restorative processes.

There is little evidence concerning CSF and serum levels of TIMP-2. CSF level of TIMP-2 was not increased in patients with meningitis complicated with epilepsy [96]. Serum TIMP-2 level did not differ between children with febrile seizures, infectious disease with fever, and GTCS [97]; however, the results were not compared with the control group of healthy children. Serum TIMP-2 increased after GTCS [16]. In our study, the serum level of TIMP-2 was increased, which might indicate its persistent neuroprotective activity; nevertheless, more studies are needed to confirm that hypothesis.

Tissue S100B level correlated with seizure frequency and loss of nerve cells in a rat model of epilepsy [98]. In a model of PTZ-induced seizures CSF, the level of S100B was increased during the first 10–30 min from seizures [99]. Increased expression of S100B was shown in patients with temporal lobe epilepsy (TLE) [22]. The serum level of S100B correlates with albumin quotient [100]. After osmotic disruption of BBB, the level of its damage measured by S100B correlated with the occurrence of seizures [101]. Increased S100B levels were observed in the serum of patients with MTLE and TLE [102,103]. Serum S100B level was increased at 30 min, 2, 6, and 24 h after GTCS [23]. However, some studies report different results, indicating no differences in S100B levels between patients with idiopathic epilepsy and control group [104] and no differences after GTCS [105]. Nevertheless, the results of a recent meta-analysis indicate that patients with epilepsy have higher serum levels of S100B [106], which is in concordance with our results and suggests chronic increase of BBB permeability in patients with epilepsy.

The data concerning serum and CSF levels of the examined proteins concentrate mainly on their level shortly after seizures. As existing data indicate that seizures influence levels of the examined proteins, there was a substantial need to fill in this gap with a carefully planned study in which the patients were examined in the interictal period and the duration of period from last seizure was observed. In this study, we tried to approach this task, which resulted in identifying MMP-9, MMP-2, TIMP-1, TIMP-2, and S100B as BBB-associated proteins whose levels are increased regardless of seizure occurrence.

### 4.2. Other Proteins Associated with BBB—ICAM-1, P-sel, CCL-2, TSP-2

In our study, levels of ICAM-1, P-sel, CCL-2, and TSP-2 did not differ between the patients with epilepsy and control group. ICAM-1 and P-sel are molecules participating in leukocyte transmigration through BBB and their expression is increased after seizures [107]. Expression of ICAM-1 is increased in patients with MTLE [108,109] and TSC [110]. Increased serum levels of ICAM-1 were observed after more than 24 h from seizures, while CSF levels of ICAM-1 remained unchanged [111]. Patients in status epilepticus also had increased serum levels of ICAM-1 and P-sel, although the study did not exclude patients with comorbidities which could affect levels of the examined proteins [31]. Studies assessing the kinetics of ICAM-1 after seizures either showed its increase after 1 h after GTCS [112] or did not show any differences [23]. Serum level of P-sel was increased at 1 h after GTCS [24]. In our study, serum levels of ICAM-1 and P-sel did not differ between the patients with epilepsy in the interictal period and the control group. It is plausible that the activation of endothelium associated with the increase of ICAM-1 and P-sel is a transient process, occurring shortly after seizures, and at that time increasing leukocyte rolling and arrest, while in the interictal period level of endothelial activation is low.

Chemokine CCL-2 (previously known as monocyte chemotactic protein-1, MCP-1) acts as a leukocyte chemoattractant [113], plays a role in leukocyte adhesion and activation [25,114], modifies ECM [115] and tight junction proteins [116], modulates synaptic transmission [117] and plasticity [118] and, by activating CCL-2/CCR2 pathway, leads to the promotion of seizures [119]. CCL-2 expression is increased in brain specimens taken from patients with MTLE, HS and FCD type II [120,121,122]. Level of CCL-2 was increased in CSF of patients with status epilepticus and did not differ between patients with chronic epilepsy and the control group [123]. Serum CCL-2 was increased in patients with HHV infection complicated with acute encephalopathy and seizures [124] and in patients with West syndrome it was either increased [125] or did not differ [126]. CCL-2 might be an important seizure-exacerbating factor by intensifying leukocyte migration, increasing BBB permeability, and tipping the pro-/anti-inflammatory balance in favor of inflammation. The fact that its serum level did not differ between the patients with epilepsy in the interictal period and control group might suggest that main activity of CCL-2 takes place during seizures and that local neuroinflammation in the interictal period might not be enough to influence serum levels of CCL-2.

TSP-2 is a molecule acting as an ECM modulator, modifying cell migration and proliferation [127], promoting excitatory synaptogenesis [128], and also non-specifically inhibiting MMPs [129]. Loss of TSP-2 was associated with the slowing of BBB repair and exacerbation of inflammation [18]. Increased activity of TSP-2 was observed in a model of cortical malformations [130] and serum levels of TSP-2 were higher in patients undergoing epilepsy surgery [19]. In our study, serum TSP-2 did not differ between the research group and the control group, which might be caused by lower disease activity in our patients. Yet, the data regarding serum TSP-2 level is scarce and more research is needed.

### 4.3. Influence of Demographic Factors on the Serum Levels of BBB-Associated Proteins

#### 4.3.1. Age

Serum levels of CCL-2 showed a weak positive correlation with age in the control group. No other BBB-associated proteins showed correlation with age in control group and patients with epilepsy. Higher serum levels of CCL-2 were observed in patients with frailty syndrome, which was considered to support the connection between CCL-2 and aging [131]. CCL-2 is suggested to promote modification of cell phenotype to the senescence-associated secretory phenotype (SASP) [132] and induce inflammaging, chronic inflammation associated with aging [133]. Positive correlation of CCL-2 with age was also shown in two studies on a healthy population [134,135]. The results of our study replicate previous findings and are in line with the theory of inflammaging.

Studies concerning correlations of serum levels of MMP-9 [136,137,138,139], MMP-2 [136,137,140], TIMP-1 [136,138,141], TIMP-2 [136,138,140] with age are either contradictory or show weak correlations with this factor. In the majority of these studies, serum S100B did not correlate with age in adult patients [32,101,142], while a negative correlation was observed in patients under 20 years of age [142]. Serum levels of P-sel [31,32,143], ICAM-1 [31,32,143,144], and TSP-2 [32,145,146] showed no correlation with age. The data suggests the lack of strong association between the serum levels of the examined proteins and age and is in line with the results of our study.

#### 4.3.2. Sex

We observed sex-dependent differences in serum levels of TIMP-2 and S100B. She serum level of TIMP-2 was higher in women both in control and in research group and the serum level of S100B was higher in men with epilepsy, while no differences were observed in the control group. Interestingly, existing data did not show differences in the serum levels of TIMP-2 of men and women [32,136,138,147]. This might suggest that factors other than sex influenced TIMP-2 level in our patients and control group, but also that a meta-analysis of a larger amount of data would be helpful to shed more light on this topic.

In our study, the serum level of S100B did not differ between women and men in the control group, which is in line with the results obtained in the literature [32,142,148,149]. Yet, in patients with epilepsy, the serum S100B level was higher in men. This finding is in contrast with previous results in which either no differences in S100B level were found between women and men with epilepsy [32,150] or higher levels of S100B were observed in women with MTLE after seizures [102]. Elevated serum S100B in men in our group of patients might be a result of different etiology of disease between the examined men and women, as there was a majority of men in the subgroup of patients with structural-metabolic epilepsy; however, a study in a larger group of patients is needed to determine possible differences in S100B level in women and men.

Both our study and previous studies examining the serum level of MMP-9 [14,136,138], MMP-2 [32,136,138], TIMP-1 [32,136,138,151], ICAM-1 [32,143], P-sel [32,143], and CCL-2 [152,153] with respect to sex-related differences did not show differences between the serum levels of these proteins in men and women. The serum level of TSP-2 in the literature either did not differ [145,146] or was higher in women [32]. Our study did not show differences in the serum TSP-2 level between women and men.

### 4.4. Correlation between the Serum Levels of BBB-Associated Proteins

We analyzed correlations between serum levels of the examined molecules and found a strong positive correlation between the serum level of MMP-9 and MMP-9/TIMP-1 ratio and between MMP-2 and the MMP-2/TIMP-2 ratio. Concurrently, TIMP-1 showed a strong negative correlation with the MMP-9/TIMP-1 ratio, but TIMP-2 showed no significant correlation with the MMP-2/TIMP-2 ratio. These results indicate that while the MMP-9/TIMP-1 ratio depends on the levels of MMP-9 and TIMP-1 in a comparable way, the MMP-2/TIMP-2 ratio is mainly influenced by serum MMP-2.

A weak negative correlation was observed between the serum level of MMP-9 and TIMP-1. MMP-9 is inhibited by TIMP-1, but it also indirectly influences TIMP-1 expression [154] via IL-1β and TNF-α [155,156]. In previous studies, a positive correlation was shown between CSF TIMP-1 and MMP-9 in patients with bacterial meningitis [96]; however, disease activity was higher in these patients and proteins were measured in CSF, which may explain the contradictory results.

The serum MMP-2 level showed a moderate positive correlation with the level of TIMP-2. A similar correlation was observed in the literature in glioblastoma cells [157]. In a study on the kinetics of serum MMP-2 and TIMP-2 after GTCS, a moderate positive correlation was observed at 72 h from seizures and weaker correlations were found at earlier time points [32]. TIMP-2 is an inhibitor of MMP-2, but it also enables activation of proMMP-2 to active MMP-2 [158]. The lack of correlation between serum level of TIMP-2 and MMP-2/TIMP-2 ratio might result from constitutive expression of TIMP-2 and less pronounced differences between the serum TIMP-2 levels in patients with epilepsy and control group.

We observed a moderate positive correlation between the serum level of MMP-9 and P-sel. A possible explanation for this correlation is MMP-9-dependent release of proinflammatory mediators, which can increase redistribution of P-sel from granules to cell surface [159]. After GTCS, the serum levels of MMP-9 and P-sel showed a weak positive correlation [32].

A moderate negative correlation was observed between the serum level of MMP-2 and TSP-2. TSP-2 is a non-specific inhibitor of MMPs [129] and its loss is associated with higher levels of MMPs [18] which may explain this correlation, especially in light of a weak negative correlation observed between serum level of TSP-2 and MMP-9. To date, only one study has analyzed the correlation between serum MMPs and TSP-2, in which no correlation was observed between serum levels of those proteins after GTCS [32]. To our knowledge, the present study is the first one that analyzes the correlation between serum levels of TSP-2 and MMPs in patients with stable epilepsy.

### 4.5. Strengths and Limitations

Our study addresses the important question of circulating proteins associated with BBB in patients with epilepsy in the interictal period. To date, most of the studies analyzed serum levels of BBB-associated proteins in a short period of time after seizures or in specific clinical situations where seizures were a co-symptom of other diseases. In these patients, changes in serum levels of BBB-associated proteins might reflect recent seizures. The design of our study focused on patients in the interictal period to avoid the influence of seizures. The period of seizure freedom was based on our previous studies [24,32]. Our study shows that serum levels of proteins associated with BBB disruption and restoration are increased in patients with epilepsy even if they have not recently experienced seizures. This observation is in line with the hypothesis of BBB dysfunction as one of the pathogenetic mechanisms of epilepsy and might indicate that serum levels of proteins associated with BBB disruption and restoration reflect chronic processes occurring at the level of BBB. A panel of BBB disruption and restoration proteins could also act as a diagnostic tool in unclear cases suspected of epilepsy.

One of the limitations of the study is its small sample size and the fact that the data are observational. Therefore, it is suggested to consider the results carefully and, whenever possible, compare them with other studies. Accordingly, we discussed our results in comparison with other studies; yet, as some of the observations are novel, it is necessary to carry out further research, preferably in larger patient groups. We selected proteins that are proven to act at the level of BBB, participating in its disruption or restoration, neuroinflammation and activation of endothelial cells, and that are clinically correlated with seizures; however, they also have many more physiological and pathological functions. We are also aware that the proteins that we evaluated may derive from peripheral vascular system and other tissues. Therefore, we designed an extensive list of exclusion criteria to minimize the impact of these factors. Additionally, blood was always collected at the same time of the day to avoid diurnal changes. Taking into consideration short half-life of some of the examined molecules, blood samples were quickly transported to the laboratory with the observance of the same procedures. Patients were advised to avoid strenuous physical activities for 24 h before blood collection to reduce the possible impact of physical exertion on molecule levels.

## 5. Conclusions

BBB dysfunction and ECM modulation are processes which are increasingly studied in the light of epilepto- and ictogenesis, yet the majority of research focuses on the impact of seizures on serum molecules. Our study addressed patients with epilepsy in the interictal period and tried to answer the question of whether serum levels of BBB-associated proteins are elevated regardless of seizure occurrence. We showed that serum levels of molecules associated with BBB disruption (MMP-2, MMP-9, S100B) and restoration (TIMP-1, TIMP-2) are increased in patients with epilepsy in the interictal period, which might reflect chronic processes taking place at the BBB, even in the absence of seizures. In our study, serum levels of molecules associated with increased activation of endothelium and leukocyte transmigration (ICAM-1, P-sel) did not differ between patients with epilepsy and the control group, which might indicate that activation of endothelial cells occurs only shortly after seizures. The serum level of CCL-2 did not differ between the research and control groups, which might suggest that the intensity of local inflammation is low. There were also no differences in the serum level of TSP-2 that might indicate that its role in chronic BBB restoration in patients with epilepsy is scarce.

The molecules whose levels are chronically elevated in serum of patients with epilepsy (MMP-2, MMP-9, S100B, TIMP-1, TIMP-2) might be considered diagnostic biomarkers of epilepsy. This finding might be useful in uncertain cases under the condition of carefully considering the factors which can influence their levels. As seizures impact serum levels of the BBB-associated molecules, we underline the importance of at least a 7-day period of seizure freedom to ascertain that seizures do not affect levels of the examined molecules in patients in the interictal period [24,32].

Our study underlines the relationship between chronic disruption and restoration of BBB and epilepsy and indicates that BBB is affected in patients with epilepsy regardless of seizure occurrence. Yet, further research in larger cohorts of patients is needed to fully understand the role of BBB in epilepsy.

## Figures and Tables

**Figure 1 cells-12-00368-f001:**
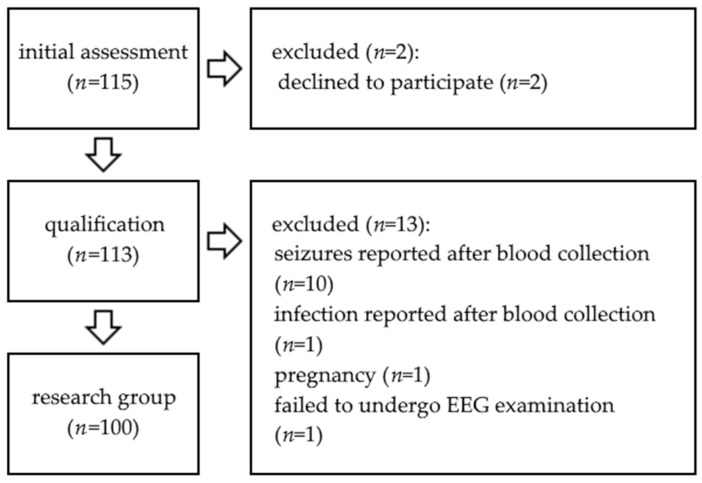
Study flow chart.

**Figure 2 cells-12-00368-f002:**
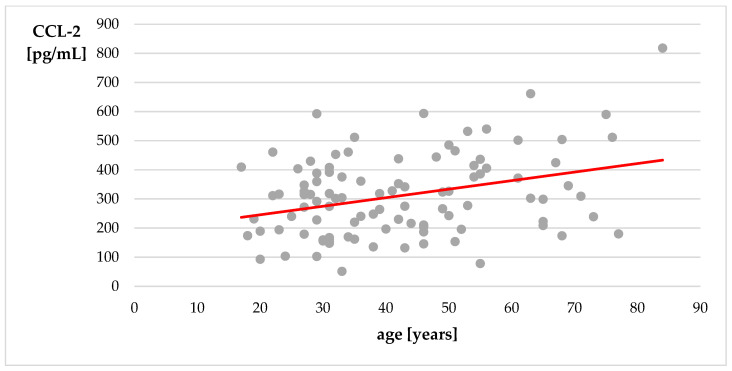
Correlation of age with level of CCL-2—control group. Spearman’s correlation coefficient.

**Figure 3 cells-12-00368-f003:**
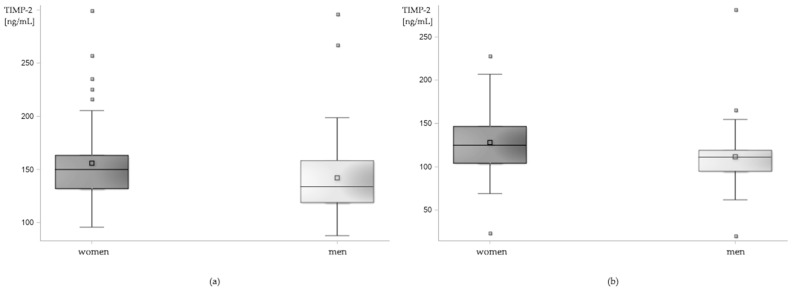
Sex-related differences in serum level of TIMP-2 in (**a**) patients with epilepsy and (**b**) control group. Box plot.

**Table 1 cells-12-00368-t001:** Inclusion and exclusion criteria.

Inclusion Criteria	Exclusion Criteria
diagnosis of epilepsy *	malignant tumor
minimum time of seizure freedom of seven days	inflammatory disease
	severe ongoing neurological disease with vascular damage(e.g., acute stroke within last six months)
	neuroimmunological disease
	immunosuppressive or immunomodulatory treatment during last six months
	surgery within last two weeks
	significant trauma within last two weeks
	hepatic, renal, or cardiac insufficiency
	severe psychiatric disease
	symptoms of infection
	CRP above the laboratory norm
	pregnancy

* on the basis of ILAE criteria from 2014 [33].

**Table 2 cells-12-00368-t002:** Demographic characteristics of research and control group.

	Research Group	Control Group
age (mean ± SEM)	43.01 ± 1.53	42.16 ± 1.58
age (median)	39	39
age (min–max)	19–82	17–84
women	52	52
men	48	48
women: age (mean ± SEM)	43.89 ± 2.17	42.65 ± 2.20
women: age (median)	39	39.5
women: age (min–max)	19–78	17–76
men: age (mean ± SEM)	42.06 ± 2.16	42.63 ± 2.30
men: age (median)	39	38.5
men: age (min–max)	22–82	20–84

**Table 3 cells-12-00368-t003:** Means and SEM values of examined proteins. Wilcoxon’s test.

	Research Group	Control Group	
	Mean ± SEM	Mean ± SEM	** *p* **
MMP-9	846.66	533.35	**<0.0001**
[ng/mL]	±56.35	±32.89	
TIMP-1	217.6	166.12	**0.001**
[ng/mL]	±11.99	±11.83	
MMP-2	294.18	200.29	**<0.0001**
[ng/mL]	±11.84	±7.00	
TIMP-2	149.29	120.12	**<0.0001**
[ng/mL]	±3.93	±3.60	
MMP-9	10.21	5.31	0.71
/TIMP-1	±2.64	±0.58	
MMP-2 /TIMP-2	1.98	1.86	**0.0087**
	±0.07	±0.14	
S100B [pg/mL]	58.93	23.62	**<0.0001**
	±9.36	±3.13	
TSP-2 [ng/mL]	29.37	33.25	0.2
	±1.75	±2.46	
ICAM-1	169.5	187.34	0.09
[ng/mL]	±6.46	±11.03	
CCL-2	333.85	310.65	0.4
[pg/mL]	±16.49	±14.09	
P-sel [ng/mL]	104.72 ± 6.62	126.82 ± 9.38	0.19

Bold font indicates statistical significance.

**Table 4 cells-12-00368-t004:** Correlations between the levels of the BBB-associated proteins. Upper line—correlation coefficient; lower line—level of significance. The values are given only for correlation coefficients ≥ 0.2. Cells are colored according to the color legend depending on the correlation coefficient.

	MMP−9	MMP−2	TIMP−1	TIMP−2	S100B	ICAM−1	P−sel	CCL−2	TSP−2	MMP−9/ TIMP−1	MMP−2/ TIMP−2
MMP−9		0.278; 0.005	−0.239; 0.016				0.408; <0.0001		−0.209; 0.037	0.722; <0.0001	0.259; 0.009
MMP−2	0.278; 0.005			0.523; <0.0001			0.254; 0.011		−0.501; <0.0001	0.218; 0.029	0.813; <0.0001
TIMP−1	−0.239; 0.016							−0.364; 0.0002		−0.797; <0.0001	
TIMP−2		0.523; <0.0001							−0.288; 0.004		
S100B											
ICAM−1											
P−sel	0.408; <0.0001	0.254; 0.011							−0.206; 0.040	0.312; 0.002	0.228; 0.023
CCL−2			−0.364; 0.0002							0.280; 0.005	
TSP−2	−0.209; 0.037	−0.501; <0.0001		−0.288; 0.004			−0.206; 0.040				−0.377; 0.0001
MMP−9/ TIMP−1	0.722; <0.0001	0.218; 0.029	−0.797; <0.0001				0.312; 0.002	0.280; 0.005			
MMP−2/ TIMP−2	0.259; 0.009	0.813; <0.0001					0.228; 0.023		−0.377; 0.0001		
correlation coefficient			positive correlation			negative correlation					
0.0–0.2											
0.2–0.4											
0.4–0.6											
0.6–0.8											
0.8–1.0											

## Data Availability

The data set can be obtained at https://docs.google.com/spreadsheets/d/1k0U_4ND0F45Wp2wqNxFSg4O4G7WUQhZg/edit?usp=share_link&ouid=108322422041402375271&rtpof=true&sd=true (accessed on 6 December 2022).

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
