# Peer review of "Blood-Brain Barrier-Associated Proteins Are Elevated in Serum of Epilepsy Patients"

_cells, 2023, doi:10.3390/cells12030368_

Round 1
Reviewer 1 Report
This study by Bronisz E et al., evaluated blood-brain-associated protein levels in the serum of epilepsy patients and controls.
Recruiting epileptic patients in their interictal period makes this manuscript (MS) interesting.
While it provided an overview of serum inflammatory markers, MS gained less attention while reading.
Below are a few minor concerns that can improve the MS quality.
Descriptive data are informative. However, what is the rationale for showing mean, min-max? Also, these values are not well discussed. I suggest simplifying the tables by providing Mean and SEM.
-Line 73: Delete 'both fulfilled inclusion criteria and did not fulfill exclusion criteria
instead, authors can simply state that Strict inclusion/exclusion criteria were applied for the recruitment of study participants or something like this.
-Table 1: Authors described the inclusion criteria while the exclusion criteria were listed in a table. For a better comparison, I would suggest adding inclusion and exclusion criteria in two columns.
-Lines 95 to 98: Investigated markers are repeated. Please reframe the sentence.
-Line 119: Authors categorized their study subjects into a Research group and a control group. However, in line 71 authors called it a ´Patient group´. In Table 3: the same group was mentioned as ´Patients with epilepsy. I suggest authors pay attention to these details.
Author Response
Dear Reviewer,
Thank you very much for reviewing our paper and providing valuable comments. We tried our best to reframe to address the comments and improve the manuscript. We hope that the revised version of the manuscript meets your high standards. We provide the point-by-point responses below. We kept the revised version of the manuscript in a review mode so that the corrections are easier to follow.
We welcome further constructive comments.
Kind regards,
Elzbieta Bronisz
1.
Descriptive data are informative. However, what is the rationale for showing mean, min-max? Also, these values are not well discussed. I suggest simplifying the tables by providing Mean and SEM.
Response 1.
Thank you for that observation. We wanted to include the additional data (i.e., min-max) to show the divergence of the obtained results. We simplified the tables.
2.
Line 73: Delete 'both fulfilled inclusion criteria and did not fulfill exclusion criteria
instead, authors can simply state that Strict inclusion/exclusion criteria were applied for the recruitment of study participants or something like this.
Response 2.
We corrected this line according to the suggestion (line 73-74).
3.
Table 1: Authors described the inclusion criteria while the exclusion criteria were listed in a table. For a better comparison, I would suggest adding inclusion and exclusion criteria in two columns.
Response 3.
We agree with that remark and we revised the table (Table 1, line 86).
Lines 95 to 98: Investigated markers are repeated. Please reframe the sentence.
Response 4.
Thank you for noticing that. We reframed the sentence to make it easier to read (lines 105-107).
5.
Line 119: Authors categorized their study subjects into a Research group and a control group. However, in line 71 authors called it a ´Patient group´. In Table 3: the same group was mentioned as ´Patients with epilepsy. I suggest authors pay attention to these details.
Response 5.
Thank you for attentive reading. We unified the nomenclature and chose to use “Research group” throughout the manuscript (throughout the text, and in Table 3), only leaving “patients with epilepsy” in few places in the Discussion where changing it for “research group” might hamper the flow of the text and hinder comparison with other studies addressing patients with epilepsy.
Reviewer 2 Report
In this manuscript, Bronisz and colleagues investigate the serum levels of molecules associated with modulation of the brain blood barrier (BBB) in patients with chronic epilepsy. By comparing patients with no seizure for at least one week with the control group (patients without epilepsy), the authors describe higher levels of molecules associated with BBB destruction and regeneration, whereas molecules associated with inflammation and immune cell recruitment are unaffected.
It is a quite linear paper but some rearrangements need to be done:
1. In the conclusion the authors mentioned that MMP-2, MMP-9, S100B, TIMP-1,TIMP-2,might be considered diagnostic biomarkers of epilepsy but, since seizure itself elevate them is the 7 days period of seizure-freedom enough to bring their level to baseline?
2. How long after blood collection were the patient followed-up? Is there a correlation between serum levels of these proteins and the time elapsed before the next seizure?
3. The variability in the control group is even higher than in the test group. The author report as control group patients without epilepsy. It is not clear which kind of patient they are. The author should confirm that the pathologies affecting these patients are not altering level of MMP2 ect as well as not altering the BBB structurally and functionally
4. Even if the numbers are quite small, could the authors find a preferential correlation with some specific epilepsy etiology?
5. The Discussion is too long and dispersive, difficult to read and follow: I suggest to move some parts to introduction and Results or re-write it in a more focused way.
Author Response
Dear Reviewer,
Thank you very much for reviewing our paper and providing valuable comments and questions. We carefully considered your suggestions and tried our best to address them to improve the manuscript. We abbreviated the part of discussion, especially concerning the background of the analyzed molecules. We provide the point-by-point responses below. We hope that the revised version of the manuscript meets your high standards.
We welcome further constructive comments. The revised version of the manuscript is in a review mode so that the corrections are easier to follow.
Kind regards,
Elzbieta Bronisz
In the conclusion the authors mentioned that MMP-2, MMP-9, S100B, TIMP-1,TIMP-2,might be considered diagnostic biomarkers of epilepsy but, since seizure itself elevate them is the 7 days period of seizure-freedom enough to bring their level to baseline?
Response 1.
We selected the period of 7-day seizure freedom based on our previous results (Cudna et al., 2016 (1), Cudna et al., 2016 (2), Cudna, 2020) and few experimental studies showing increased BBB markers for about 7 days after the induced seizures. In this previous research we observed the kinetics of MMP-2, MMP-9, S100B, TIMP-1, and TIMP-2 after generalized tonic-clonic seizures and found that serum levels of these proteins were increased after seizures (MMP-9, TIMP-1, TIMP-2, S100B, and a tendency for higher level, but without statistical significance in case of MMP-2) but at the time point of 72 hours after seizures their level did not differ from control group (TIMP-2, MMP-2, S100B) or there was a trend for lowering the protein concentration (MMP-9, TIMP-1) to the value before seizures by day 4 after seizures. In the light of these results we decided to analyze our patients at least 7 days after seizures. Additionally, the median of seizure freedom was 140 days, and 35% of patients did not have seizures during last year.
Cudna A, Jopowicz A, Mierzejewski P, Kurkowska-Jastrzębska I. Serum metalloproteinase 9 levels increase after generalized tonic-clonic seizures. Epilepsy Res. 2017 Jan;129:33-36. doi: 10.1016/j.eplepsyres.2016.11.006.
Cudna, A.; Jopowicz, A.; Bronisz, E.; Kurkowska-Jastrzębska, I. Blood-brain barrier markers after tonic-clonic seizures. In Proceedings of the 12th European Congress on Epileptology; Prague, Czech Republic, (11-15. September 2016).
Cudna, A. Activation of blood-brain barrier in epilepsy - dynamics of the expression of blood-brain barrier markers after seizures. Doctoral thesis, Institute of Psychiatry and Neurology, Warsaw, 2020.
How long after blood collection were the patient followed-up? Is there a correlation between serum levels of these proteins and the time elapsed before the next seizure?
Response 2.
We followed up the patients for one year. Additionally, we followed up a group of 50 patients for another year and we designed a study examining the influence of the serum levels of the analyzed proteins on seizure count where we showed that MMP-9, MMP-2, and CCL-2 influence the seizure count during the follow-up (Bronisz et al., 2022). Unfortunately, we did not collect the data on the dates of the next seizure after examination, but maybe our other study could be of interest.
Bronisz, E.; Cudna, A.; Wierzbicka, A.; Kurkowska-Jastrzębska, I. Serum Proteins Associated with Blood–Brain Barrier as Potential Biomarkers for Seizure Prediction. Int. J. Mol. Sci. 2022, 23, doi:10.3390/ijms232314712.
The variability in the control group is even higher than in the test group. The author report as control group patients without epilepsy. It is not clear which kind of patient they are. The author should confirm that the pathologies affecting these patients are not altering level of MMP2 ect as well as not altering the BBB structurally and functionally
Response 3.
The control group consisted of mostly healthy volunteers and subjects hospitalized due to noninfectious diseases, with no history of infection or seizures. We took great care to exclude any subjects who had diseases or states that could influence the levels of the analyzed proteins and we used the same list of exclusion criteria for both research and control group. We added that statement in Material and Methods (line 78).
Even if the numbers are quite small, could the authors find a preferential correlation with some specific epilepsy etiology?
Response 4.
It is a very good question and yes, we tried to find the differences between the subgroups of different etiologies. We found that in the group of patients with genetic epilepsy serum level of TSP-2 was higher while in patients with structural-metabolic epilepsy higher were serum levels of MMP-2, MMP-9, and P-selectin. We refrained from adding this data as it would inevitably make the Discussion excessively long which would hamper its reading. Yet we plan to publish the data on the correlation of serum levels of the analyzed proteins with the epilepsy etiology separately. I hope it would be an interesting addition to the current manuscript.
The Discussion is too long and dispersive, difficult to read and follow: I suggest to move some parts to introduction and Results or re-write it in a more focused way.
Response 5.
Thank you for the remark. As the introduction was marked as providing sufficient background, we believed the better option was to re-write the Discussion and because of that we revised it, trying to make it more concise and easier to follow. We revised the Discussion and merged chapters 4.1.2. Proteins Associated with BBB Disruption and Restoration in Animal Models and Surgical Specimens and 4.1.3. Proteins Associated with BBB Disruption and Restoration in CSF and Serum into one chapter: 4.1.2. Proteins Associated with BBB Disruption and Restoration in Epilepsy. We tried our best to make the Discussion more concise without losing the information necessary to follow our reasoning, yet we feel that we cannot leave the large amount of data presented in the manuscript uncommented. We divided the Discussion into subchapters to facilitate reading.
Round 2
Reviewer 1 Report
I appreciate the authors for revising the manuscript. I have no further comments.